# Transmissible cancers and the evolution of sex under the Red Queen hypothesis

**Thomas G. Aubier** ⊙*, **Matthias Galipaud, E. Yagmur Erten** ⊙, **Hanna Kokko** ⊙

Department of Evolutionary Biology and Environmental Studies, University of Zurich, Zurich, Switzerland

* thomas.aubier@normalesup.org

**Data Availability Statement:** The computer code of the simulations and of the analyses (R, version 3.4.4) is provided as a Supporting Information file 'S1 Source Code'.

## Abstract

The predominance of sexual reproduction in eukaryotes remains paradoxical in evolutionary theory. Of the hypotheses proposed to resolve this paradox, the 'Red Queen hypothesis' emphasises the potential of antagonistic interactions to cause fluctuating selection, which favours the evolution and maintenance of sex. Whereas empirical and theoretical developments have focused on host-parasite interactions, the premises of the Red Queen theory apply equally well to any type of antagonistic interactions. Recently, it has been suggested that early multicellular organisms with basic anticancer defences were presumably plagued by antagonistic interactions with transmissible cancers and that this could have played a pivotal role in the evolution of sex. Here, we dissect this argument using a population genetic model. One fundamental aspect distinguishing transmissible cancers from other parasites is the continual production of cancerous cell lines from hosts' own tissues. We show that this influx dampens fluctuating selection and therefore makes the evolution of sex more difficult than in standard Red Queen models. Although coevolutionary cycling can remain sufficient to select for sex under some parameter regions of our model, we show that the size of those regions shrinks once we account for epidemiological constraints. Altogether, our results suggest that horizontal transmission of cancerous cells is unlikely to cause fluctuating selection favouring sexual reproduction. Nonetheless, we confirm that vertical transmission of cancerous cells can promote the evolution of sex through a separate mechanism, known as similarity selection, that does not depend on coevolutionary fluctuations.

## Introduction

Sexual reproduction entails several and often severe costs [1], yet most eukaryotes engage in sex, at least occasionally [2]. To explain this apparent paradox, much theory has been developed to identify the benefits associated with sexual reproduction [3–6]. Sex shuffles genetic material from parent individuals and breaks apart allele combinations built by past selection. Whether this is selected for depends strongly on the stability of the environment. In a stable environment, selection is likely to have already brought favourable combinations of alleles together in the past, and continuing genetic mixing can become deleterious [7, 8]. In many models, therefore, the evolution of sex relies on the advantage that lineages receive from

**Funding:** This research was supported by grants from the Swiss National Science Foundation to HK (310030B_182836; http://www.snf.ch/en/Pages/default.aspx), from the National Institutes of Health to EYE and HK (U54 CA217376; https://www.nih.gov/), and from the University Research Priority Program (URPP) "Evolution in Action" of the University of Zurich to EYE and HK (https://www.evolution.uzh.ch/en.html). The funders had no role in study design, data collection and analysis, decision to publish, or preparation of the manuscript.

**Competing interests:** The authors have declared that no competing interests exist.

mixing genetic materials in an environment that is not stable [9]. Under a scenario of fluctuating selection, sex can be beneficial because it breaks apart allele combinations that have been built by past selection and that by now have become disadvantageous [8, 10].

The 'Red Queen hypothesis' for the evolution of sex emphasises the potential of host-parasite interactions to cause fluctuating selection, thus favouring genetic mixing [11–17] (not to be confused with the macroevolutionary Red Queen hypothesis [18]). Hosts with a rare genotype are often less susceptible to common parasite strains [19], and the resulting coevolution between hosts and parasites (so-called 'Red Queen dynamics') yields negative frequency-dependent selection, such that selection fluctuates over time for host and parasite alike [20]. Genetic mixing is then favoured because it can produce offspring with genetic associations that are currently rare and therefore less susceptible to parasites [14, 20, 21].

Until now, studies of the Red Queen hypothesis have considered hosts and parasites that belong to distinct taxonomic groups. In a recent opinion piece, Thomas and colleagues (2019) break from this tradition and highlight the intriguing potential role of transmissible cancerous cell lineages for the evolution of sex [22]. Cancer manifests itself as somatic cells breaking free of multicellular cooperation and proliferating uncontrollably, often at the cost of the organism's (as well as the cancer cells') life [23]. Cancer is an evolutionary dead end [24], but an exception arises when cancer cells are transmissible and outlive their host, behaving in this respect identically to parasites that infect new hosts. In this latter case, hosts can suffer not only from cancer cells arising from their own tissues but also from transmitted cancer cells that originated in other host individuals.

Transmissible cancers are fundamentally different from contagious agents that elevate cancer risk (e.g., human papillomavirus causing cervical cancers [25]). Transmissible cancers are directly transmitted to new hosts and do not require cells or viral particles of another taxon to play any role in the infection system. So far, transmissible cancerous cell lines have been observed in a few taxa only, namely three mammal species [26–29] and four bivalve species [30–32], but early multicellular organisms, with presumably basic anticancer defences, may have been plagued by this problem more than extant ones [22, 33].

On this basis, Thomas and colleagues (2019) propose the intriguing hypothesis that the prevalence of sex in multicellular eukaryotes may have been originally driven by transmissible cancerous cell lines regularly infecting multicellular hosts (note, however, that eukaryotes tend to be sexual even if they are unicellular) [22]. According to this view, the diversity of genotypes created by sex helps individuals in the task of differentiating between self and non-self, thus reducing susceptibility to transmissible cancers. Selection on the multicellular host to avoid infection by transmissible cancer is therefore akin to selection induced by heterospecific parasitic agents, favouring the evolution and maintenance of sex.

Transmissible cancers are indeed similar to other parasites in that the long-term survival of their lineages depends on their successful transmission to other hosts, which requires circumventing the host's immune system. Transmissible cancer cells, however, present a particularly thorny problem for the host, as their genetic makeup (and hence their cellular phenotype) is by default very similar to that of the host, since they originate via mutation from the host's own tissue or from a conspecific tissue (note, however, that a case of cross-species transmission has been reported [31]). Self/non-self recognition is conceptually similar to the 'matching-alleles' model of host defence against parasites (relevant for the Red Queen process [20, 34–37]), in that the cancer cell's infection prospects depend on the genotypic composition of the host population. At first glance, this sets the scene for antagonistic coevolution between the transmissible cancer and its host, favouring the evolution of sex just as predicted under the Red Queen hypothesis. Without formal inquiries, however, it is difficult to judge whether antagonistic interactions between hosts and transmissible cancers lead to fluctuating selection

of the type that is essential for Red Queen dynamics to take place. Specifically, it is unclear whether fluctuating selection can be maintained when susceptible hosts themselves produce the parasitic cancerous lines to which they are susceptible.

In this paper, we investigate whether antagonistic interactions between hosts and transmissible cancers can promote the evolution of sex under the Red Queen hypothesis. We analyse a population genetic model of fluctuating selection and complement it with an epidemiological model. The latter model builds an explicit epidemiological setting that we then use to examine the likely parameter values that the population genetic model takes. This combined use of two models allows us to evaluate how likely it is for the modelled system to find itself within a selection regime in which Red Queen dynamics can favour sexual reproduction.

Our approach is an intentionally simplified version of all self/non-self recognition systems (with only two loci involved in recognition, plus one modifier that determines sexual/asexual reproduction), and we regard it as the first necessary step for understanding the conditions under which interactions between hosts and transmissible cancers can yield to Red Queen dynamics promoting the evolution of sex. We focus on one of the fundamental aspects that distinguish transmissible cancers from other parasites: the continual production of cancerous cell lines from hosts' tissues (a process that we refer to as 'neoplasia'), which we show to inhibit the evolution of sex under the Red Queen hypothesis. The inhibitory effect arises because 'neocancers' produced via neoplasia are likely to infect hosts with a common genotype, strongly reducing the lag between hosts and transmissible cancers evolutionary dynamics, which is necessary for coevolutionary fluctuations to occur. Coevolutionary dynamics can select for sex under some parameter regions of the population genetic model, yet we show that the size of those regions shrinks once we account for epidemiological constraints.

## Models and results

### Population genetic model

We extend a standard population genetic model of the Red Queen hypothesis [38–41] to account for neoplasia, i.e., the fact that cancers originate from conspecific hosts and bring their genotypes into the population of transmissible cancer cells. We distinguish between two stages that characterise transmissible cancer cells: cancer cells successfully transmitted to a new host in a previous generation (called 'transmitted cancers'), and those that do not yet have such an infection history but are directly derived from the original host where neoplasia occurred (called 'neocancers'). Neocancers become transmitted cancers as soon as they successfully infect a new host. We specifically test whether coevolution between hosts and transmissible cancers can favour the evolution of sex under the Red Queen hypothesis.

We follow the genotypic frequencies of haploid hosts and haploid cancer cells through a life cycle that consists of a census, reproduction, neoplasia (development of neocancers), and selection (that depends on interactions between hosts and transmissible cancers). We assume that hosts and cancer cells each form populations of sufficient size such that we can ignore the effects of genetic drift. We also assume that time is discrete.

**Genotypes.**   Hosts and transmissible cancers are haploid and have two loci, *A* and *B*, with two possible alleles (A/a, B/b) that determine the outcome of the interaction between hosts and cancers.

Hosts possess an additional modifier locus *M* with two possible alleles (M/m) that determine whether the host reproduces sexually or asexually (here, gene order of *A*, *B* and *M* does not matter). Hosts carrying the allele M reproduce sexually, whereas hosts carrying the allele m reproduce asexually. Although cancer cells are ultimately derived from host cells and thus carry the modifier locus *M* too, we assume for simplicity that they never engage in sex and

recombination; consequently, there is no need to distinguish between cancer cells with alleles m and M in the model. Our approach does not give cancer cells themselves the ability to fuse and recombine [42, 43], as this allows us to follow the original argumentation of Thomas and colleagues (2019) [22].

As a whole, therefore, hosts are of eight possible genotypes (mAB, mAb, maB, mab, MAB, MAb, MaB, Mab) whose frequencies are $f_i^{\mathrm{H}}$, and transmissible cancer cells are of four possible genotypes (AB, Ab, aB, ab) whose frequencies are $f_j^{\mathrm{C}}$.

**Reproduction.** Host individuals carrying the allele M at the modifier locus *M* engage in sexual reproduction. Random mating (which brings two haploid genomes together) is followed immediately by meiosis. During meiosis, loci *A* and *B* recombine at rate $r_{\mathrm{host}}$. Recombination between these loci (*A* and *B*) and locus *M* is irrelevant, since sexual progeny inherit the allele M at locus *M* from both parents. By contrast, host individuals carrying allele m at locus *M* engage in asexual reproduction, and genotypic frequencies in the asexual host population do not change. All transmissible cancer cells reproduce clonally; i.e., genotypic frequencies in transmissible cancer cells do not change.

Mutations at the interaction loci occur at a rate of $m_{\mathrm{host}}$ and $m_{\mathrm{cancer}}$ per locus per generation in the hosts and the cancer cells, respectively. If a mutation occurs, allele A (respectively, B) becomes allele a (respectively, b), and vice versa. We assume there is no mutation at the modifier locus *M*, since our aim is to assess whether an allele M controlling sexual reproduction can invade the population once it is introduced.

Overall, from frequencies $f_i^{\mathrm{H}}$ and $f_j^{\mathrm{C}}$, we can calculate the genotypic frequencies $f_i'^{\mathrm{H}}$ and $f_j'^{\mathrm{C}}$ in hosts and cancer cells after reproduction, assuming that generations do not overlap.

**Host neoplasia and development of neocancers.** Transmissible cancers are ultimately derived from neoplasia, and have initially the same genotype as the neocancer's original host (i.e., we assume no somatic mutation/selection during the oncogenic process that produces the cancer within the original host). We use $\alpha$ to denote the proportion of transmissible cancers that are 'neocancers' (without infection history), with genotypic frequencies $f_j'^{\mathrm{H}}$ at the interaction loci. The remaining proportion $1-\alpha$ of cancers are 'transmitted cancers' (with infection history), with genotypic frequencies $f_j'^{\mathrm{C}}$ differing from those of neocancers. If $\alpha = 0$, neoplasia does not occur, and transmissible cancers are, in that case, best seen as classical heterospecific parasites that do not arise from host cells themselves.

Therefore, after neoplasia (the development of neocancers) has occurred, genotypic frequencies in the transmissible cancers are:

$$f_j''^{\mathrm{C}} = \alpha f_j'^{\mathrm{H}} + (1 - \alpha)f_j'^{\mathrm{C}} \tag{1}$$

We assume that neoplasia does not depend on host genotype—i.e., hosts of any genotype develop neocancers at the same rate. We also assume that host genotype does not impact the severity of fitness consequences (to the host) of neoplasia. Therefore, even if neoplasia increases host mortality, this does not change genotypic frequencies in the host population: $f_i''^{\mathrm{H}} = f_i'^{\mathrm{H}}$.

**Selection.** During the selection phase, hosts are assumed to encounter transmissible cancers (neocancers and transmitted cancers) proportionally to their frequency. Changes in genotypic frequencies are determined by the match/mismatch at the interaction loci (*A* and *B*).

We implement the commonly used matching-alleles interaction model, now interpreted in the context of self/non-self recognition. When there is an exact match between genotypes *i* and *j* of the host and the infecting cancer, the infecting cancer has high fitness (which we model as fitness coefficient $w_{i,j}^{\mathrm{C}} = 1$), whereas the host suffers a fitness cost (fitness coefficient $w_{i,j}^{\mathrm{H}} = 1 - s_{\mathrm{host}}$). In the alternative case where the genotypes *i* and *j* of the host and the infecting

cancer do not match, the host retains its high fitness advantage (fitness coefficient $w_{i,j}^{\mathrm{H}} = 1$), whereas the infecting cancer suffers a fitness cost (fitness coefficient $w_{i,j}^{\mathrm{C}} = 1 - s_{\mathrm{cancer}}$).

Assuming that the probability of interaction between host genotype $i$ and cancer genotype $j$ is the product of their respective frequencies, $f_i''^{\mathrm{H}}$ and $f_j''^{\mathrm{C}}$, the frequency of host genotype $i$ after selection is $\tilde{f}_i^{\mathrm{H}} = \dfrac{f_i''^{\mathrm{H}} . w_i^H}{\sum_k f_k''^{\mathrm{H}} . w_k^H}$, where $w_i^H$ is the fitness of host genotype $i$, given by $w_i^H = \sum_j f_j''^{\mathrm{C}} w_{i,j}^{\mathrm{H}}$. The genotypic frequencies in cancer after selection, $\tilde{f}_j^{\mathrm{C}}$, are calculated analogously (with fitness coefficients $w_{i,j}^{\mathrm{C}}$).

During the selection process presented above, we therefore made the assumption that transmission occurs only horizontally. In S2 Appendix, however, we implement vertical transmission of cancerous cells from mother to offspring. Vertical transmission causes 'similarity selection' that has also the potential to favour sex and recombination, while being distinct from genotypic selection created by coevolutionary fluctuations in standard Red Queen models [44]. Agrawal (2006) used a standard population genetic approach to emphasise the importance of similarity selection for the evolution of sex [44], but more theoretical and empirical work is required to confirm that this is a potent mechanism favouring sexual reproduction. To our knowledge, the only other models considering similarity selection in other evolutionary contexts are those of Greenspoon and M'Gonigle [45, 46].

**Numerical simulations.** We initiate the populations assuming that all hosts are asexual (in hosts, the frequency of allele m at locus $M$ is set to 1), and other allele frequencies are initialized randomly; i.e., genotypic frequencies are drawn following uniform distributions over the range [0,1] and are then normalized such that their sum is equal to 1 in hosts and transmissible cancers.

The initial host and cancer populations are allowed to coevolve for 10,000 generations (burn-in period), with the dynamics computed using the recursion equations above. The mutant allele M is then introduced such that 5% of the host population becomes sexual (switches from allele m to allele M), and host and cancer populations are allowed to coevolve for 1,000,000 generations. For each combination of parameters tested, we perform 100 simulations characterised by different initial conditions, such that coevolutionary dynamics can explore different limit cycling dynamics. If we find the frequency of allele M to reach and maintain a frequency >0.999 for at least 500 generations over the course of at least one simulation, we assume that sex can invade under the combination of parameters implemented.

To provide a sensitivity analysis, we vary the parameters $\alpha$ (proportion of neocancers among the population of transmissible cancers) and ($s_{\mathrm{host}}$,$s_{\mathrm{cancer}}$) (selection coefficient associated with the interaction between hosts and cancers), with mutation rates $m_{\mathrm{host}} = m_{\mathrm{cancer}} = 10^{-5}$. For each combination of parameters, we run simulations with different values of $r_{\mathrm{host}}$ $\in\{0.005, 0.01, 0.02, 0.05, 0.1, 0.2, 0.3, 0.4, 0.5\}$ (recombination rate if hosts reproduce sexually). If sex can invade for at least one value of $r_{\mathrm{host}}$, then we consider that sex can evolve under the combination of parameters ($\alpha$,$s_{\mathrm{host}}$,$s_{\mathrm{cancer}}$) tested. Note that our assumption of implementing $\alpha$ as a parameter is a simplification that we make to aid conceptual understanding of the role that newly arisen cancers play in Red Queen dynamics; we thereafter switch to viewing $\alpha$ as an emergent property of the system in the epidemiological model presented and analysed below. The computer code of the simulations and of the analyses is provided as S1 Source Code.

## Red Queen dynamics and evolution of sex

Under the Red Queen hypothesis, antagonistic interactions between hosts and parasites cause fluctuating selection leading to non-steady coevolutionary dynamics (Red Queen dynamics)

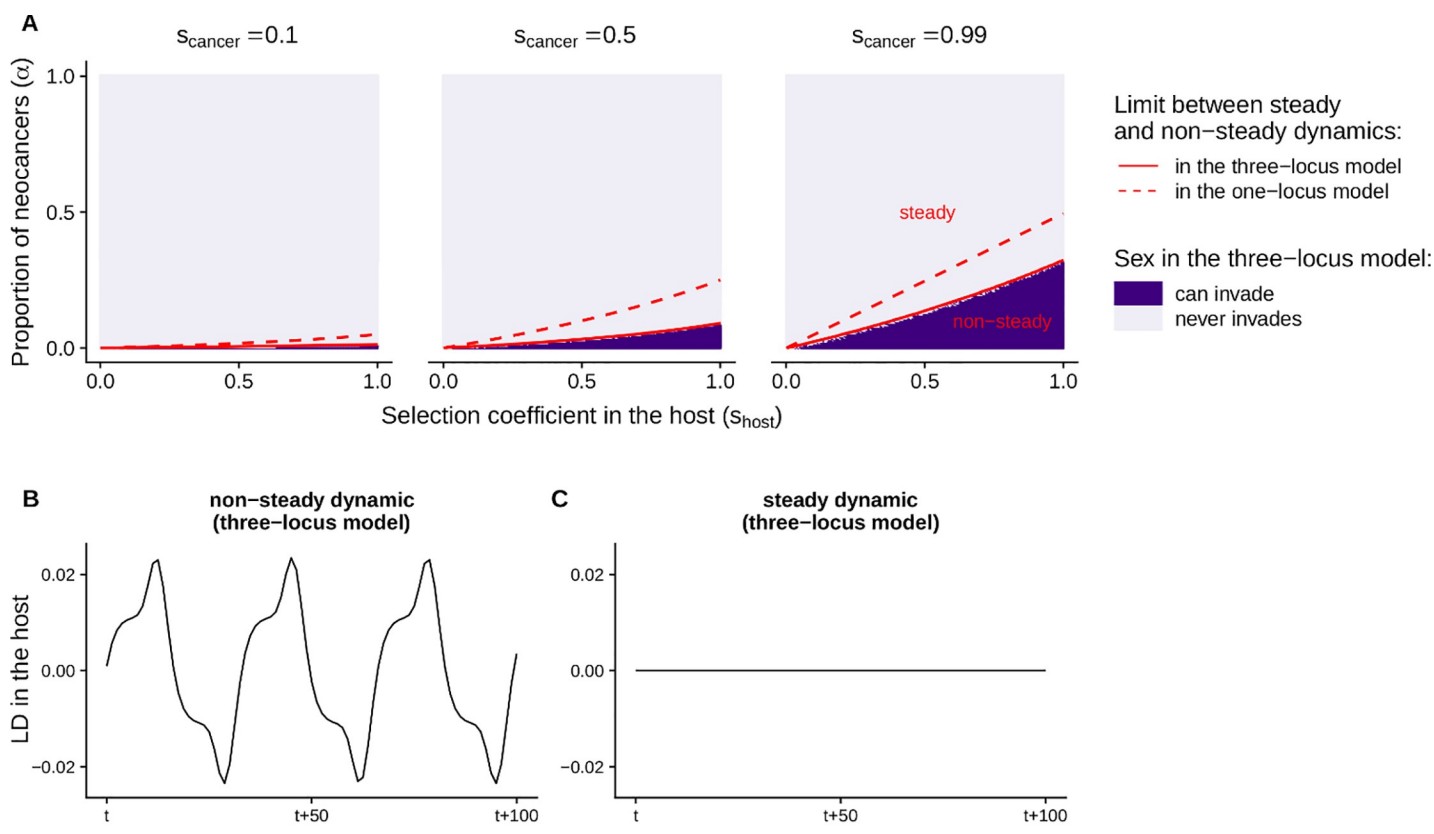

**Fig 1. Coevolutionary dynamics between hosts and transmissible cancers, and evolution of sex. (A)** Sensitivity of the population genetic models to the selection coefficients ($s_{host}$, $s_{cancer}$) and to the proportion of transmissible neocancers that are recently derived from the original host ($\alpha$). Red lines delimit the parameter spaces leading to non-steady and steady coevolutionary dynamics in the three-locus model (plain line; found numerically) and in the simplified one-locus model (dashed line; found analytically in S1 Appendix). In the three-locus model, the dynamic is defined as 'steady' when the variance in genotypic frequencies over 500 time steps is below $10^{-10}$. Dark purple indicates conditions under which a modifier allele associated with sexual reproduction (and with recombination, at least for one of the recombination rates tested) can invade in the three-locus model in at least one of the 100 simulation runs. The code used to perform this sensitivity analysis can be found in S1 Source Code. **(B-C)** Examples of non-steady and steady coevolutionary dynamics in the three-locus model. The linkage disequilibrium in the host is calculated as $LD = f^H_{ab}f^H_{AB} - f^H_{Ab}f^H_{aB}$, i.e., a positive linkage disequilibrium here represents a non-random excess of allele combinations ab and AB. In (B) and (C), parameter values are: $s_{host} = 0.5$, $s_{cancer} = 0.8$, and $\alpha = 0$ (B) or $\alpha = 0.1$ (C).

that favours the evolution of sex. Without neoplasia ($\alpha = 0$), we show that such dynamics occur in the our three-locus population genetic model (cf. parameter spaces delimited by the plain red lines in Fig 1A; e.g., Fig 1B). Consequently, sexual reproduction can invade in the host population (in purple in Fig 1A), especially if it associates with an intermediate recombination rate (S1 Fig). These predictions are in agreement with previous theory on the Red Queen hypothesis.

The results change when some of the transmissible cancers in circulation are neocancers, the results of neoplasia occurring in the original hosts ($\alpha > 0$). Even a small proportion of neocancers is sufficient to bring the coevolutionary dynamics between hosts and transmissible cancers to a halt (Fig 1A; e.g., Fig 1C). Neoplasia tightens the link between genotypic frequencies in host and cancer (Eq 1), in the sense of reducing any time lag between the two evolutionary dynamics. Since a time lag is necessary for coevolutionary fluctuations to occur and for sex to be favoured in Red Queen models, the continual production of new cancer cells from original hosts makes it impossible for sexual reproduction to invade if the proportion of neocancers is too high (Fig 1A).

Other strong determinants of the coevolutionary dynamics are the strengths of selection associated with the interaction between hosts and transmissible cancers ($s_{host}$,$s_{cancer}$). Overall, increased strengths of selection promote non-steady coevolutionary dynamics and favour the evolution of sex (Fig 1A). In other words, when resistance or infection associate with a high fitness in host and cancer respectively, Red Queen dynamics can occur and favours the evolution of sex.

Notably, we get the same results when we consider that the ancestral reproductive mode is facultative sexual reproduction (with hosts originally reproducing sexually and asexually to the same extent; S2 Fig). Neoplasia also dampens coevolutionary cycling when we consider more than two interaction loci mediating the interaction between hosts and cancers (S3 Fig); note that increasing the genotypic space constrains the conditions under which sex can evolve under the Red Queen hypothesis (as shown in [38]).

To gain further insights into the effect of the proportion of neocancers ($\alpha$) on coevolutionary dynamics between hosts and transmissible cancers, we consider a simplified one-locus population genetic model and solve it analytically. This model is based on a single autosomal haploid locus $A$ with two possible alleles (A/a), controlling the interaction between hosts and transmissible cancers. This model does not include a modifier locus controlling the reproduction mode of the host; all hosts are reproducing asexually. We also assume that there is no mutation ($m_{host} = m_{cancer} = 0$). We determine the local stability of all equilibria by analysing the eigenvalues of the corresponding Jacobian matrices (S1 Appendix). We show that all equilibria are unstable, leading to non-steady coevolutionary dynamics, only if the proportion of neocancers is lower than a threshold value $A^*(s_{host},s_{cancer})$:

$$\alpha < A^*(s_{host},s_{cancer}) \tag{2}$$

with:

$$A^*(s_{host},s_{cancer}) = \frac{s_{host}s_{cancer}}{(2 - s_{host})(2 - s_{cancer}) + s_{host}s_{cancer}} \tag{3}$$

This condition is represented by a dashed red line in Fig 1A. The maximum value of $\alpha$ that allows for a non-steady coevolutionary dynamic becomes higher as selection coefficients increase ($\frac{\partial A^*}{\partial s_{host}} > 0$ and $\frac{\partial A^*}{\partial s_{cancer}} > 0$), as found numerically in the model with two interaction loci (plain red line in Fig 1A). This analytical derivation reveals that the amplitude of fluctuations in selection is not merely getting small, but is deterministically shrinking to zero as $\alpha$ increases. Therefore, in this simplified genetic setting, neoplasia always dampens the Red Queen dynamics. This explains why in the three-locus model, sex cannot evolve if the proportion of transmissible cancers deriving from host neoplasia is too high.

Thus far, we have considered horizontal transmission only, which in our model fails to promote the evolution of sex as soon as neoplasia dampens coevolutionary fluctuations. Although the theory on the Red Queen hypothesis relies on non-steady coevolutionary dynamics, antagonistic interactions can favour the evolution of sexual reproduction via other processes. In S2 Appendix, we show that vertical transmission of cancerous cells can promote the evolution of sex through a separate mechanism, called similarity selection [44], in which the sex-promoting effect operates in the absence of coevolutionary fluctuations. Similarity selection occurs when there is a cost to being genotypically similar to one's family members. In particular, because of the transmission of cancerous cells from parent to offspring, this cost exists if infection compatibility is under genetic influence (e.g., as in the matching-alleles system of our population genetic model).

## Epidemiological model

The population genetic model presented above treats the proportion of transmissible neocancers originating via neoplasia ($\alpha$) and the selection coefficient caused by infection by transmissible cancers ($s_{\text{host}}$) as independent parameters. Yet these two parameters are likely to be linked in a more realistic epidemiological context, and some combinations of ($\alpha$, $s_{\text{host}}$) may be more likely than others. We therefore build an explicit epidemiological model to determine the epidemiological settings that may favour the evolution of sex (as predicted by the population genetic model, above).

In our epidemiological model, host individuals can be cancer-free ('susceptible'), develop a neocancer (through neoplasia), or be infected by a transmitted cancer (see the flowchart of the model, Fig 2). Notably, infected hosts incur an elevated mortality rate. We do not model the possibility of a host having both types of cancers simultaneously, but include a parameter ($\theta$) that allows infection status to change from one type to another. Thus, $\theta$ controls whether neoplasia can 'take over' if the host has a transmitted cancer already and whether, conversely, a host undergoing neoplasia can become infected by another cancer.

Since the purpose of the model is to investigate likely values of $\alpha$ depending on $s_{\text{host}}$, rather than track any Red Queen dynamics, we do not include any variance in hosts' and parasites' genotypes. Instead, we assume that no host can recognise a transmissible cancer as non-self and fend it off; for the sake of analytical tractability, we thus underestimate the proportion of neocancers and overestimate the selection coefficient, by making it easy for cancers to continue infecting hosts beyond the original one (as shown in S4 Appendix).

The following ordinary differential equations control the changes in densities of susceptible hosts ($S$), hosts that developed a neocancer by neoplasia ($I_0$), and hosts that are infected by a transmitted cancer ($I_T$):

$$\begin{cases} \dfrac{\mathrm{d}S}{\mathrm{d}t} = & bN\left(1 - \dfrac{N}{K}\right) - \left(\mu + \lambda_0 + \beta\dfrac{I_0 + I_T}{N}\right)S \\[2ex] \dfrac{\mathrm{d}I_0}{\mathrm{d}t} = & \lambda_0(S + \theta I_T) - \left(\mu + v + \theta\beta\dfrac{I_0 + I_T}{N}\right)I_0 \\[2ex] \dfrac{\mathrm{d}I_T}{\mathrm{d}t} = & \beta\dfrac{I_0 + I_T}{N}(S + \theta I_0) - (\mu + v + \theta\lambda_0)I_T \end{cases} \qquad (4)$$

with $N = S + I_0 + I_T$.

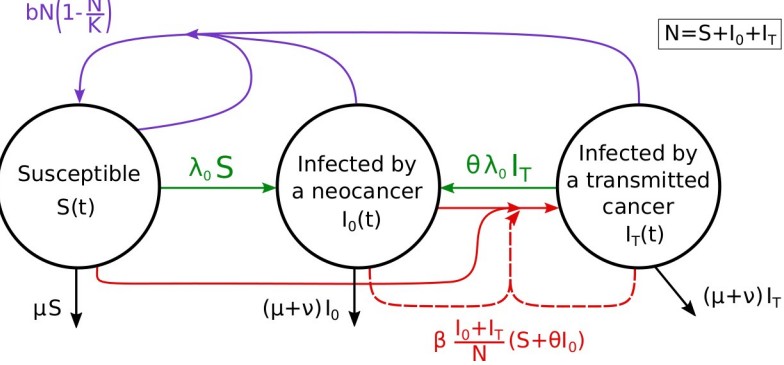

**Fig 2. Flowchart of the epidemiological model.**

The host birth rate is density-dependent with a baseline birth rate $b$ and a carrying capacity $K$. Baseline mortality rate, independent of density or infection status, is $\mu$, which is elevated to $\mu+v$ in infected hosts (thus, $v$ denotes the additional host mortality caused by the cancer). Neoplasia makes hosts develop neocancers at a rate $\lambda_0$, and hosts can become infected by a transmitted cancer, controlled by a rate $\beta$ and dependent on the prevalence of transmissible cancers. Parameter $\theta$ controls the change in infection status. If $\theta = 0$, one individual can only ever host one type of cancer. If $\theta > 0$, hosts can change infection status.

## Epidemiological equilibrium

**Prevalence of transmissible cancers.** Depending on the parameter values, the host population either goes extinct or persists with $S^* + I_0^* + I_T^* > 0$ at equilibrium (S3 Appendix). This equilibrium state is stable and features an endemic infection by transmissible cancer ($I_0^* > 0$ and $I_T^* > 0$; S3 Appendix). Note that an equilibrium where all hosts are susceptible, which is possible in standard epidemiological models (SI, SIR, and SIS models), is not a feature of our model because susceptible individuals continually produce neocancers.

At equilibrium, we can derive the expression of the prevalence $P^*$ of transmissible cancers, defined as $\frac{I_0^* + I_T^*}{S^* + I_0^* + I_T^*}$ (S3 Appendix; see also S4 Fig):

$$P^*(\lambda_0, \beta, \mu, v) = \frac{\beta - \lambda_0 - \mu - v + \sqrt{(\beta - \lambda_0 - \mu - v)^2 + 4\beta\lambda_0}}{2\beta} \tag{5}$$

**Proportion of neocancers.** At equilibrium, we determine the expression of the proportion of neocancers ($\hat{\alpha}$) that we define as $\frac{I_0^*}{I_0^* + I_T^*}$ (S3 Appendix):

$$\hat{\alpha} = \frac{\lambda_0}{\mu + v}\left[\frac{1}{P^*(\lambda_0, \beta, \mu, v)} - 1\right] \tag{6}$$

**Strength of selection due to transmissible cancers.** We also determine the expression of the selection coefficient caused by transmissible cancers ($\hat{s}_{\text{host}}$) that we define as the mean life span reduction due to the risk of being infected by transmissible cancers (S3 Appendix):

$$\hat{s}_{\text{host}} = \frac{1}{1 + \frac{\mu}{v}\frac{1}{P^*(\lambda_0, \beta, \mu, v)}} \tag{7}$$

## Epidemiological setting favouring the evolution of sex

To infer the epidemiological conditions that can favour the evolution of sex, we compare the properties of the equilibrium state ($\hat{\alpha}, \hat{s}_{\text{host}}$) to the conditions that favour the evolution of sex in the previous three-locus population genetic model, assuming the approximations $\hat{\alpha} \approx \alpha$ and $\hat{s}_{\text{host}} \approx s_{\text{host}}$.

Some parameters—the birth rate ($b$), the carrying capacity ($K$), and the changes in infection status ($\theta$)—have no effect on the proportion of neocancers ($\hat{\alpha}$) and on the strength of selection caused by transmissible cancers ($\hat{s}_{\text{host}}$) at the equilibrium (those parameters do, however, affect the dynamics of our epidemiological model before reaching the equilibrium state). By contrast, other parameters—$\lambda_0$, $\beta$, $\mu$, and $v$—affect $\hat{\alpha}$ and $\hat{s}_{\text{host}}$ either directly or via their effect on the prevalence of transmissible cancers ($P^*$) (Eqs 6 and 7; Figs 3 and 4).

**Rate of neoplasia ($\lambda_0$).** A high rate of neoplasia associates with a high proportion of neocancers ($\hat{\alpha}$) at equilibrium (Figs 3 and 4A) and with strong selection caused by transmissible

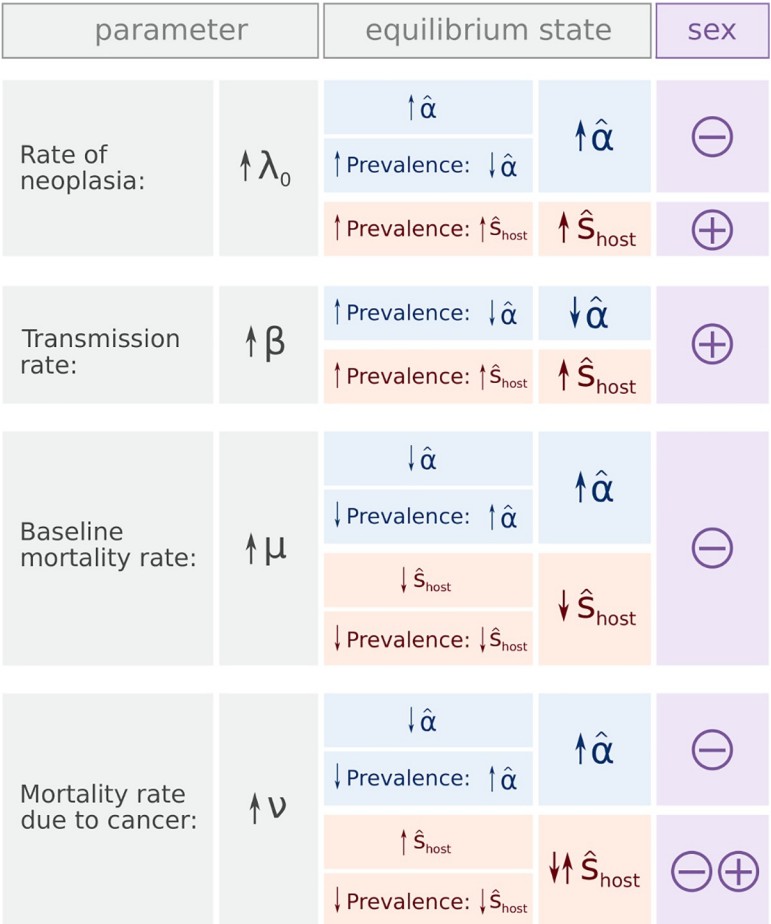

**Fig 3. Sensitivity to epidemiological parameters, based on the signs of partial derivatives.** Each epidemiological parameter can affect the proportion of neocancers ($\hat{\alpha}$; in blue) and the strength of selection caused by transmissible cancer ($\hat{s}_{host}$; in red) at equilibrium, either directly or via its effect on prevalence of transmissible cancers (cf. Eqs 6 and 7; and see S3 Appendix). The right panel under the column 'equilibrium state' denotes the overall effect of a change in parameter value on $\hat{\alpha}$ or $\hat{s}_{host}$. From these sensitivity analyses, we infer whether changes in the epidemiological settings can favour the evolution of sex or not (as predicted in the previous population genetic model; in purple). Note that $\hat{\alpha}$ and $\hat{s}_{host}$ do not depend on parameters $b$, $K$, and $\theta$.

cancers ($\hat{s}_{host}$) (via high cancer prevalence; Figs 3 and 4B). Although strong selection might favour sex, the high proportion of neocancers counteracts this effect, and our population genetic model predicts that this epidemiological setting as a whole does not favour the evolution of sex (Fig 1).

On the other hand, a low rate of neoplasia associates with a low proportion of neocancers at equilibrium, which on its own would favour sex, but this also associates with transmissible cancers only causing weak selection. Here, weak selection drives the result that sex is, once again, not favoured (Fig 1). Epidemiological considerations therefore constrain the conditions under which sex evolves: it is difficult to combine a low proportion of neocancers with strong selection in the host (S5 Fig). Overall, a low rate of neoplasia is more likely to lead to conditions favouring the evolution of sex than a high rate of neoplasia (Fig 4C).

**Transmission rate of cancers ($\beta$).** A high transmission rate of cancers associates with a low proportion of neocancers at equilibrium (Figs 3 and 4A) and with strong selection caused by transmissible cancers (via high cancer prevalence; Figs 3 and 4B) at equilibrium. As a result,

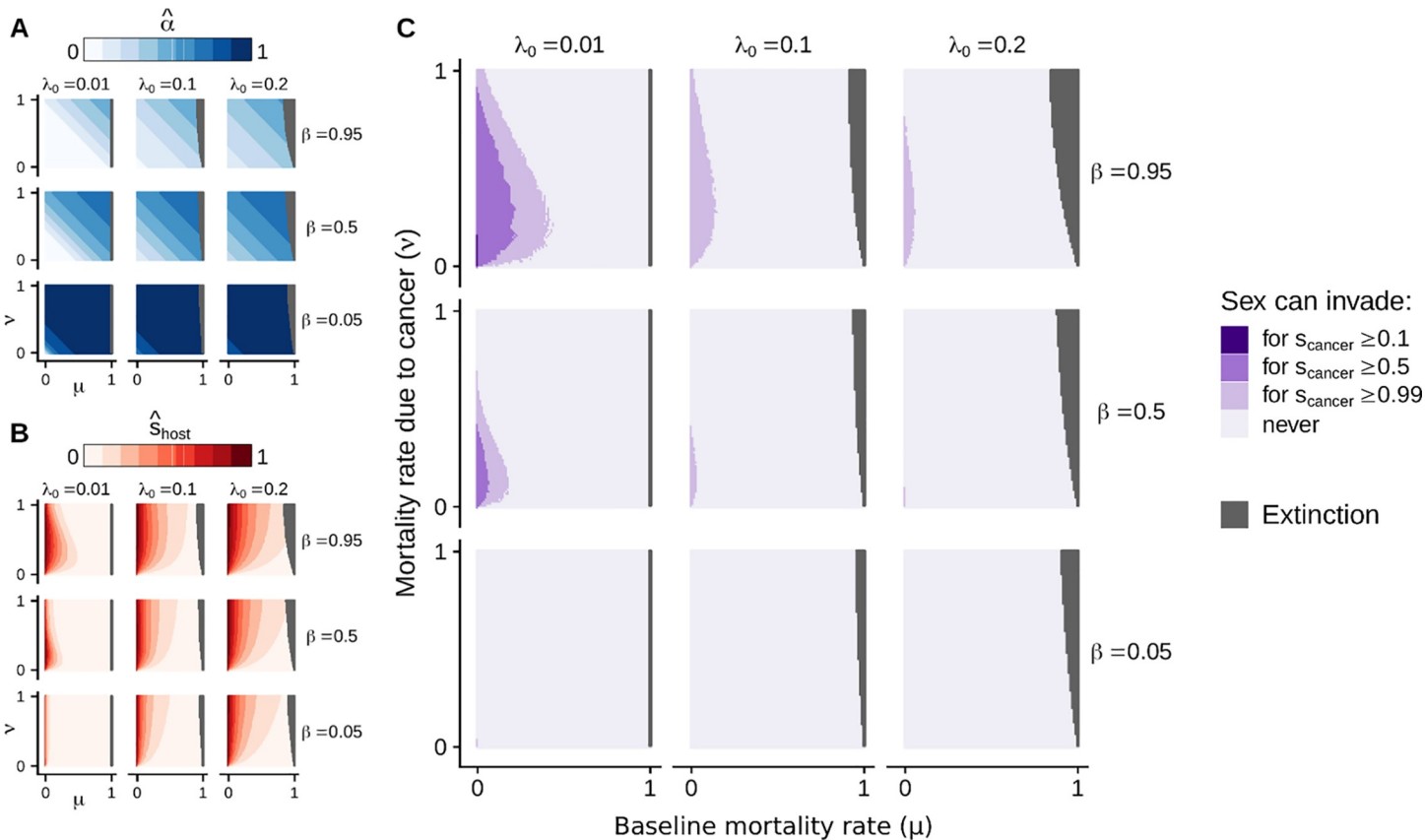

**Fig 4. Sensitivity to epidemiological parameters, and conditions favouring sex. (A, B)** Effects of epidemiological parameters on $(\hat{\alpha}, \hat{s}_{host})$ at equilibrium. **(C)** Conditions under which sex should be favoured (inferred from the three-locus population genetic model; Fig 1A). In grey, we represent the conditions under which the host population gets extinct, assuming that the baseline birth rate $b$ equals to one (condition leading to extinction: $\mu + vP^*(\lambda_0, \beta, \mu, v) > b$; see S3 Appendix). The code used to perform this sensitivity analysis can be found in S1 Source Code.

a high transmission rate of cancers offers favourable conditions for the evolution of sex (Figs 3 and 4C).

**Mortality rates ($\mu$ and $v$).** High mortality rates associate with a high proportion of neo-cancers at equilibrium (Figs 3 and 4A). This effect occurs in our model because infected individuals dying at a rate $\mu + v$ are replaced by susceptible newborn individuals (flow represented in purple in Fig 2), and high mortality rates therefore associate with low cancer prevalence. The consequent low cancer prevalence reduces the risk of becoming infected by a transmitted cancer, leading to a high proportion of neocancers at equilibrium.

For high mortality rates, the consequent low cancer prevalence weakens selection caused by transmissible cancers (Eq 7). Simultaneously, however, the relative mortality rate due to cancer ($v/\mu$) directly strengthens selection caused by transmissible cancers (Eq 7). Therefore, a high baseline mortality rate ($\mu$) directly weakens selection caused by transmissible cancers, in addition to its indirect effect via cancer prevalence. The situation is different in the case of a high cancer-associated mortality rate ($v$), which has opposite direct and indirect effects on selection caused by transmissible cancers. Indeed, the high relative mortality rate due to cancer ($v/\mu$) directly strengthens selection, whereas the consequent low cancer prevalence weakens selection. Analytical derivations (S3 Appendix) show that for a high baseline mortality rate ($\mu > \beta - \lambda_0$), a high cancer-induced mortality rate (high $v$) leads to strong selection. For a low baseline

mortality rate ($\mu \leq \beta - \lambda_0$), selection caused by transmissible cancers is instead maximised at intermediate values of cancer-induced mortality (intermediate $v$) (Fig 4B).

As a result, the combination of a low baseline mortality rate $\mu$ and an intermediate cancer-associated mortality rate $v$ leads to conditions favouring the evolution of sex (Fig 4C).

As a whole, our results suggest that the conditions for transmissible cancers to select for sex are most favourable if transmissible cancers exist in a situation with a low rate of neoplasia (low $\lambda_0$), a high transmission rate (high $\beta$), and an intermediate cancer-associated mortality rate (intermediate $v$) in hosts with a low baseline mortality rate (low $\mu$). Our epidemiological model highlights that the proportion of neocancers ($\alpha$) and the selection coefficient caused by transmissible cancers ($s_{host}$), used as independent parameters in the population genetic model, become easily linked when viewed in an epidemiological setting. Put more precisely, epidemiology constrains the conditions leading to both a low proportion of neocancers and a high selection in host (S5 Fig), because the prevalence of transmissible cancers impacts both parameters simultaneously. Our results therefore suggest that the range of parameter values providing the conditions under which sex can invade is relatively narrow (Fig 4C). The evolution of sex is even less likely as selection on transmissible cancer due to failed infection ($s_{cancer}$) decreases, regardless of other parameter values (Fig 4C).

## Discussion

At the early stages of multicellularity, when anticancer defences were presumably less developed or prevalent than today, organisms may have been under considerable risk of transmissible cancers. At first glance, transmissible cancerous lines, akin to parasites, could induce selection on multicellular hosts to avoid infection, causing fluctuating selection and favouring the evolution of sex (Red Queen hypothesis). Nonetheless, transmissible cancers differ fundamentally from parasites: transmissible cancers are a priori genetically similar to their hosts because they originally derive from hosts' tissues. Our formal theoretical investigation shows that antagonistic interactions between multicellular organisms and transmissible cancerous lines only rarely lead to fluctuating selection when we account for this fundamental aspect. As a result, although transmissible cancers may favour the evolution of sex in multicellular hosts under the Red Queen hypothesis, they only appear to do so under very restricted conditions.

Specifically, we investigated the implication of neoplasia—i.e., the production of cancerous lines from multicellular hosts' tissues—for the evolution of sex under the Red Queen hypothesis. If one genotype is common in the host population, neoplasia mostly produces transmissible cancerous lines that can successfully infect this common genotype. Nonetheless, we find a striking effect of neoplasia on the coevolutionary dynamic. Neoplasia tightens the coupling between genotypic frequencies in multicellular hosts and transmissible cancers, which reduces the lag between hosts and transmissible cancers evolutionary dynamics, and thereby inhibits coevolutionary fluctuations. As the coevolutionary system reaches a stable polymorphism, fluctuating selection vanishes and sex becomes deleterious.

Our reasoning highlights that transmissible cancers share features with other processes that have been shown to inhibit the evolution of sex by dampening coevolutionary fluctuations. These include overlapping generations [47] (but see [48]) and epidemiological dynamics (i.e., changes in parasite prevalence [49–51]), explaining the absence of coevolutionary cycling when we considered two types of transmissible cancers in our epidemiological model (S4 Appendix). Notably, parasite transmission occurring mostly among genetically similar hosts can also diminish coevolutionary cycling. It can do so by allowing the parasite population to

respond more quickly to frequency changes in the hosts [45, 52], an effect similar to what we found for neoplasia.

In our population genetic model, the strength of selection associated with the interaction between hosts and transmissible cancers proves to be an additional important determinant of the long-term coevolutionary dynamics (see [10] for similar results in a standard Red Queen model). Despite the dampening effect of neoplasia, coevolutionary fluctuations may persist when host-cancer interactions associate with strong enough selection. Our population genetic model treats the proportion of transmissible cancers that recently derived from neoplasia and the selection coefficient in the host as independent parameters (following an 'open' approach [53]). Our complementary model with an explicit epidemiological context (a 'closed' approach [53]), however, shows that the values of the parameters are likely to covary in a predictable manner. Neoplasia increases the prevalence of transmissible cancers, which strengthens selection imposed by transmissible cancers. Therefore, although a low rate of neoplasia on its own would enhance coevolutionary cycling, it tends to associate with low cancer prevalence, which has the opposite effect: it weakens selection and dampens coevolutionary cycling. Since it is difficult to find conditions under which both parameters take values that enhance fluctuating selection, the epidemiology of transmissible cancers as a whole constrains the conditions under which sex can evolve under the Red Queen hypothesis.

Under our main model assumptions, sexual reproduction evolves as a defensive strategy against transmissible cancers only under very restricted conditions. A slow host life history, a low rate of neoplasia, a high transmission rate, and an intermediate virulence synergistically provide the best-case scenario for fluctuating coevolutionary dynamics favouring the evolution of sex. So far, horizontally transmitted cancers have been identified in very few species in the wild, namely in dog (canine transmissible venereal tumour), in Tasmanian devil (devil facial tumour diseases), and in four bivalve species (clam leukaemia) [33]. In each case, phylogenetic analyses revealed that few monophyletic lines of transmissible cancerous cells widely spread in populations [27, 29, 32, 54], suggesting low rates of neoplasia and high transmission rates. In two of the three cases, infection by contemporary transmissible cancers is highly virulent (in bivalve and Tasmanian devil [55, 56]). According to our model predictions, the epidemiology of known contemporary transmissible cancers therefore matches (at least qualitatively) the restricted conditions prone to the evolution and maintenance of sex. Unfortunately, inferring the epidemiology of early transmissible cancers from those contemporary cases is speculative at best, because the nature of early transmissible cancers may have been very different when anticancer defences were only beginning to evolve [22, 57]. Notably, sexual reproduction is ancestral in dogs, devils, and bivalves, and as an evolutionary innovation it also precedes multicellularity [2, 58, 59].

Since the focus of our model was to analyse how neoplasia affects antagonistic coevolutionary dynamics, this may come at a cost of ignoring other key features of transmissible cancers. In the following paragraphs, we therefore discuss how relaxing our modelling assumptions may change the predictions.

The life history and physiology of hosts appear to determine the epidemiology of contemporary transmissible cancers (a result somewhat akin to general ideas about cancer in short- and long-lived organisms [60]). Whereas cancer cells are able to live freely outside of their host in bivalves [61], the transmission of cancer cells requires physical contact between hosts in mammals (in Tasmanian devil and in dog [28]). In particular, transmission mainly occurs via social contacts (fights and biting) or, in the case of dogs (where the tumours occur in genitalia), during copulation. Assuming that transmission requires close physical proximity among hosts, and assuming that sex cannot occur without it (a statement that excludes broadcast spawning for instance) while asexual reproduction can, then the transmission rate among sexual individuals

would be particularly high [1]. Similarly, although skin is generally an effective barrier against immunogenic agents, sexual behaviour may expose less protected parts of the body, enabling transmissions of cancer cells (as in the case of canine venereal tumours [28]). Our model did not implement any direct differences in infection dynamics between sexuals and asexuals brought about by the physics of mating. We considered a best-case scenario for the evolution of sex under the Red Queen hypothesis, in which the only cost associated with sexual reproduction is the recombination load—i.e., the risk of breaking apart beneficial combinations of alleles. Although not explicitly modelled, additional costs would presumably constrain even further the conditions under which transmissible cancers can promote the evolution of sex.

Contemporary transmissible cancers have emerged and spread in a manner that allows us to track the sequence of evolutionary changes. In dog and Tasmanian devil, transmissible cancers appear to have derived from populations with low genetic diversity [62–64]. More precisely, in Tasmanian devil, a transmissible cancer (Devil Facial Tumour 2) has been shown to express the host's major histocompatibility complex class 1 (MHC1) molecules and to match alleles that are either most widespread or non-polymorphic in the host population [65]. Simultaneously, however, an independent but older transmissible cancer (Devil Facial Tumour 1) avoids the immune response by down-regulating the expression of the MHC1 molecules [66]. Therefore, in mammals, low genetic diversity might allow newly derived transmissible cancers to persist longer in host populations, giving them time to acquire mutations necessary to infect genetically dissimilar hosts (e.g., by down-regulating their immune response). This is not a process taken into account by our model. Then again, invertebrates somewhat differ from this picture, as they do not possess a vertebrate-like adaptive immune system or a major histocompatibility complex, which vertebrates utilize for discriminating self from non-self. Consequently, in bivalves, transmissible cancers seem to easily infect genetically dissimilar hosts, as illustrated by cases of cross-species transmissions [31, 32].

For simplicity, our model ignores any details of how hosts reject cancers as non-self apart from a simple matching-alleles interaction among host and transmissible cancer. Real-life examples of Red Queen dynamics can be genetically rather complex: in a system of *Bacillus thuringiensis* infecting nematodes *Caenorhabditis elegans*, coevolution appears to involve copy number variations (which can evolve very rapidly) in the pathogen but many loci with small effect in the host [67]. As our model was not tailored to any particular system, our matching-alleles interaction model simplifies away any system-specific detail, and uses version of a self/non-self recognition system that is known to favour coevolutionary cycling in models examining the Red Queen hypothesis outside the realm of cancer [20, 34–37]. To examine more diverse multilocus settings appears a worthwhile avenue for further work. Choosing the most appropriate genetic architecture is, as a whole, a challenge because we do not know much about the allorecognition systems of the earliest multicellular organisms. It appears important to remember that if these mechanisms were incomplete (or absent), then organisms would be susceptible to transmissible cancers regardless of their genetic matching with infectious cancerous cells. This scenario would appear to inhibit coevolutionary fluctuations, and the evolution of sexual reproduction would, once again, not have involved the Red Queen hypothesis between hosts and cancers. However, whether this verbal argument holds for incipient (and therefore imprecise) self/non-self detection mechanisms is presently unclear. In any case, the transition to multicellularity appears to have involved regulatory changes that still play a role in cancer [68], but whether there is also a link to sex is uncertain.

In line with the literature on the Red Queen hypothesis, we focused on the implication of fluctuating selection dynamics for the evolution of sex. Nonetheless, any antagonistic interactions can also cause 'arm race' dynamics characterised by the continual accumulation of adaptive mutations [69]. In such context of arm race dynamics, sexual reproduction can be

beneficial by speeding up adaptation [70]. In the face of infection by transmissible cancers, sex could accelerate the development of immunity to cancerous cell lineages. More generally, by focusing on the standard Red Queen hypothesis, we did not model other evolutionary processes that can favour the evolution of sexual reproduction (except similarity selection via vertical transmission; as discussed below).

In our main analysis, we assumed that transmissible cancers could infect any host in the population (horizontal transmission, assuming random mixing of hosts and parasites). In an additional analysis, we explored the implications of vertical transmission of cancerous cells from parent to offspring for the evolution of sexual reproduction (as advocated by Thomas and colleagues 2019 [22], and as suggested by the poor recognition of fetal cells during pregnancy in mice when embryos are identical to their mother [71]), and we confirmed that this mode of transmission involves a separate mechanism, similarity selection, which can promote the evolution of sex without relying on coevolutionary fluctuations [44]. Very few studies have reported instances of vertical transmission of cancerous cells [72]. In humans, for instance, only few cases of in utero cancer transmissions from mother to fetus are known [73, 74]. Altogether, our results show that horizontal and vertical transmission lead to different pathways to sex. Whether the latter option still leads to sex when increasing the realism of our first model remains to be seen: we did not, for example, consider that cancer-ridden individuals might be poor at reproducing, which will reduce the prevalence of vertically transmitted cancers in the population. Threats such as late-life neocancer or infection by a horizontally transmitted cancer may also select for other traits than sex. Life histories with risks of poor late-life performance may select for early reproduction without necessarily involving a change in reproductive mode (sexual or asexual) (as shown in the Tasmanian devil [75]). Although general theory for condition-dependent sex exists (and cancer, by leading to poor condition, could conceivably promote such patterns) [76–79], the relative likelihood of different responses remains to be investigated theoretically.

In the metazoan hydra, asexual reproduction ('budding') can associate with direct vertical transmission of cancer cell, in which the division of 'parental' cancer cells contribute to the production of newly 'budded' multicellular offspring [72]. Interestingly, sexual reproduction often associates with the production of unicellular zygotes, which has been suggested to prevent the vertical transmission of noncooperative (cancerous) cells in multicellular organisms [80, 81]. In brief, unicellular bottlenecks might represent an efficient way of exposing noncooperative elements to selection early in development, hence limiting their propagation in the population. Future theoretical works could fruitfully test to what extent the transition to multicellular life, and the subsequent risk of transmitting cancer cells vertically, may have promoted the evolution of unicellular bottlenecks and of sexual reproduction (e.g., with a mechanistic model as in [82, 83]). This could potentially provide a more plausible mechanism than fluctuating selection for favouring the evolution or maintenance of sexual reproduction under high cancer risk.

Overall, our theoretical models suggest that antagonistic interactions between early multicellular organisms and transmissible cancers favour the evolution of sexual reproduction as predicted under the Red Queen hypothesis only under restricted conditions. While infection by transmissible cancers causes negative frequency-dependent selection in the multicellular host, neoplasia dampens fluctuating selection (and therefore cycling coevolutionary dynamics), which underpins the Red Queen hypothesis for the evolution of sex. Nonetheless, infection by transmissible cancers could have favoured sexual reproduction via other evolutionary processes. In particular, we confirm that similarity selection caused by vertical transmission of cancerous cells could favour the evolution of sexual reproduction even without fluctuating selection.

## Supporting information

**S1 Fig. Evolution of sex associated to different recombination rates ($r_{host}$).** Sensitivity of the three-locus population genetic model to the selection coefficients ($s_{host}$,$s_{cancer}$) and to the proportion of neocancers that are recently derived from the original host ($\alpha$). The conditions under which sex can invade are more restricted when sex associates with a high recombination rate (leading to a high recombination load). Sex without genetic mixing is neutral compared to asexual reproduction (i.e., if $r_{host} = 0$ in our haploid case, not shown). Therefore, sex is more strongly favoured if it associates with an intermediate recombination rate $r_{host}$. See Fig 1 for more details.
(TIF)

**S2 Fig. Evolution from facultative sex (50% asexual reproduction and 50% sexual reproduction) to obligate sex (100% sexual reproduction).** Sensitivity of the population genetic models to the selection coefficients ($s_{host}$,$s_{cancer}$) and to the proportion of transmissible neocancers that are recently derived from the original host ($\alpha$). Red lines delimit the parameter spaces leading to non-steady and steady coevolutionary dynamics. The dynamic is defined as 'steady' when the variance in genotypic frequencies over 500 time steps is below $10^{-10}$. Dark purple indicates conditions under which a modifier allele associated with obligate sexual reproduction (and with recombination, at least for one of the recombination rates tested) can invade in at least one of the 100 simulation runs. We get the same results as in Fig 1.
(TIF)

**S3 Fig. Evolution of sex in population genetic models with three or four interaction loci.** Sensitivity of the population genetic models to the selection coefficients ($s_{host}$,$s_{cancer}$) and to the proportion of transmissible neocancers that are recently derived from the original host ($\alpha$). Red lines delimit the parameter spaces leading to non-steady and steady coevolutionary dynamics. The dynamic is defined as 'steady' when the variance in genotypic frequencies over 500 time steps is below $10^{-10}$. Dark purple indicates conditions under which a modifier allele associated with sexual reproduction (and with recombination, at least for one of the recombination rates tested) can invade in at least one of the 100 simulation runs. Sex is favoured mostly within a restricted genotypic space under the Red Queen hypothesis (as shown in [38]; but see [84] and [85] accounting for other evolutionary processes favouring sexual reproduction). Additionally, neoplasia dampens coevolutionary cycling even when considering more than two interaction loci.
(TIF)

**S4 Fig. Prevalence $P^*$ of transmissible cancers at equilibrium.** A high prevalence associates with a high selection coefficient in the host (high $\hat{s}_{host}$, Fig 4B). In grey, we represent the conditions under which the host population gets extinct, assuming that the baseline birth rate $b$ equals to one (condition leading to extinction: $\mu+\nu P^*(\lambda_0,\beta,\mu,\nu)>b$; see S3 Appendix).
(TIF)

**S5 Fig. Conditions under which sex can invade (inferred from the three-locus population genetic model) in the epidemiological model.** At equilibrium, we determine the values of $(\hat{\alpha},\hat{s}_{host})$, and the conditions favouring the evolution of sex are inferred (A) from $\hat{\alpha}$ only or (B) from $(\hat{\alpha},\hat{s}_{host})$. Decreased rate of neoplasia ($\lambda_0$) leads to the conditions of $\hat{\alpha}$ that are prone to the evolution of sex (A). Nonetheless, it also associates with a decrease in the selection coefficient caused by transmissible cancers (Fig 4B), thereby inhibiting the evolution of sex (B). In grey, we represent the conditions under which the host population gets extinct, assuming that the baseline birth rate $b$ equals to one (condition leading to extinction: $\mu+\nu P^*(\lambda_0,\beta,\mu,\nu)>b$; see S3

Appendix). See Fig 1 for more details.
(TIF)

**S1 Appendix. Analytical derivation of a simplified one-locus population genetic model.**
(PDF)

**S2 Appendix. Numerical analysis of a population genetic model with similarity selection.**
(PDF)

**S3 Appendix. Analytical derivation of the epidemiological model.**
(PDF)

**S4 Appendix. Numerical analysis of an epidemiological model with two types of cancers.**
(PDF)

**S1 Source Code. Simulation code (R, version 3.4.4).**
(ZIP)

## Acknowledgments

This study was initiated as a follow-up to exciting discussions at the Kokkonuts EcoEvo journal club. We thank Joanna Masel, Frédéric Thomas and one anonymous reviewer for comments that have helped improve our manuscript.

## Author Contributions

**Conceptualization:** Thomas G. Aubier, Matthias Galipaud, E. Yagmur Erten, Hanna Kokko.

**Formal analysis:** Thomas G. Aubier.

**Investigation:** Thomas G. Aubier, Matthias Galipaud.

**Methodology:** Thomas G. Aubier.

**Supervision:** Hanna Kokko.

**Validation:** Thomas G. Aubier, Matthias Galipaud, E. Yagmur Erten, Hanna Kokko.

**Visualization:** Thomas G. Aubier.

**Writing – original draft:** Thomas G. Aubier, Matthias Galipaud, E. Yagmur Erten.

**Writing – review & editing:** Thomas G. Aubier, Matthias Galipaud, E. Yagmur Erten, Hanna Kokko.

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
