## [Editor Report · Decision Letter 0]

5 Apr 2020

Dear Dr Aubier, 

Thank you for submitting your manuscript entitled "Transmissible cancers do not easily select for sexual reproduction" for consideration as a Research Article by PLOS Biology.

Your manuscript has now been evaluated by the PLOS Biology editorial staff as well as by an academic editor with relevant expertise and I am writing to let you know that we would like to send your submission out for external peer review.

Please re-submit your manuscript within two working days, i.e. by Apr 07 2020 11:59PM.

Kind regards,

Di Jiang,

Associate Editor

PLOS Biology

---

## [Decision Letter · Decision Letter 1]

11 May 2020

Dear Dr Aubier,

Thank you very much for submitting your manuscript "Transmissible cancers do not easily select for sexual reproduction" for consideration as a Research Article at PLOS Biology. Your manuscript has been evaluated by the PLOS Biology editors, an Academic Editor with relevant expertise, and by three independent reviewers.

In light of the reviews (below), we will welcome re-submission of a revised version that takes into account the reviewers' comments. Our Academic Editor agrees with reviewer 1 that you should ideally consider an open-ended model where many loci co-evolve; however, as this could involve extensive additional work, we have decided to make it optional but not essential. Our Academic Editor suggests that the revision should consider vertical transmission (which should not be hard) and also at least discuss how facultative sex might evolve towards obligate sex. We cannot make any decision about publication until we have seen the revised manuscript and your response to the reviewers' comments. Your revised manuscript is also likely to be sent for further evaluation by the reviewers.

We expect to receive your revised manuscript within 2 months. 

**IMPORTANT - SUBMITTING YOUR REVISION**

*Re-submission Checklist*

*Published Peer Review*

*PLOS Data Policy*

*Blot and Gel Data Policy*

Sincerely,

Di Jiang, PhD

Associate Editor

PLOS Biology

REVIEWS:

Reviewer #1 (Joanna Masel, signed review): This manuscript is an extended rebuttal of the novel hypothesis recently proposed in an essay by Thomas et al. in PLoS Biology, that protection from transmissible cancer drove the evolution of obligate sexuality as animals evolved to become multicellular. I agree with the authors of the current manuscript that the proposal of Thomas et al. should have been accompanied by a mathematical demonstration of proof of principle. However, the current manuscript attempts to provide a disproof of principle, which is always a trickier affair, as changing any number of assumptions might change the outcome, and proof of principle requires only one plausible scenario to be found.

My first major concern is with a modeling assumption that is widespread within the host-parasite coevolution theory community, but utterly baffling to someone like myself outside that who comes in grounded in molecular evolution data. This assumption is that only a small number of loci (two in this manuscript) mediate the host-parasite interaction, that fluctuating selection dynamics occur within this extremely restricted genotype space, and that it is lags within these fluctuations that drive the advantage of host sex in evolving away from parasites. The alternative "arms race" dynamic (see eg doi:10.1111/j.1420-9101.2008.01598.x for discussion) envisages open-ended evolution in many more loci, without the return to previous genotypes that characterizes fluctuations in more restricted genotype spaces that have nowhere else to go. As an outsider, I do not understand why fluctuating selection dynamics have received so much more theoretical attention than arms races; molecular studies of host-parasite interactions show adaptation occurring across vast tracts of the host genome (see eg the work of David Enard). In the context of this manuscript, it seems to me that a much larger genotype space, such that sexual progeny are almost always different from their parents, would be closer to Fig. 1B of Thomas et al., and must be considered before their hypothesis is dismissed.

My second major concern is that my reading of Thomas et al. is that their hypothesis applies to the transition from facultatively sexual reproduction to obligately sexual reproduction. While Thomas et al. is in places sloppily written on this point (another reason that a formal model should have been required in that work to make it clearer), I nevertheless think this is still the only reasonable reading. But the current manuscript models the transition from zero sex to obligate sex, even while it makes critical remarks about sex being an ancestral trait. It would be more appropriate to model the transition from rare sex to obligate sex.

Because of these two major concerns, I do not find that the principle proposed by Thomas et al. has been satisfactorily disproved in this work. Major revisions to address these two concerns has the potential to change the conclusions of this work. Minor revisions to acknowledge without addressing the concerns would substantially reduce the impact of the work.

Signed,

Joanna Masel

Reviewer #2 (Frederic Thomas, signed review): Comments of the paper: "Transmissible cancers do not easily select for sexual reproduction", by Thomas G. Aubier, Matthias Galipaud, E. Yagmur Erten, Hanna Kokko

Few months ago, I proposed (Thomas et al. 2019) with few colleagues in a verbal paper a novel hypothesis to explain the evolution of sexual reproduction. Our hypothesis suggested that transmissible cancer cells could have played, and would still play, a major role for the evolution and maintenance of sexual reproduction in the living world. In this paper, Aubier et al. explored the conditions that are required for this hypothesis to be plausible. Their main conclusion is that the conditions for transmissible cancers to select for sex are most favorable if transmissible cancers exist in a situation with a low rate of neoplasia, a high transmission rate and an intermediate cancer-associated mortality rate in hosts with a low baseline mortality rate. Because these conditions are very particular, they conclude that transmissible cancers do not easily select for sexual reproduction.

I found this paper excellent, timely and well written. I am also glad to see that our hypothesis had stimulated other scientists, and yielded beautiful work like this one. 

I suggested Minor Revisions, but in fact it is between minor and major, I think. 

As authors acknowledge themselves at the end of their ms, they only considered horizontal transmission while we also suggested in Thomas et al. 2019 that vertical transmission could exert a major role, even a higher role than horizontal one. Thus, the correction I would ask, both to be fair and also to stimulate other complementary researches, is in their title. In my view, it should be something like : "Horizontal transmission of malignant cells does not easily select for sexual reproduction" (instead of Transmissible cancers do not easily select for sexual reproduction). I would REALLY appreciate if the authors could change their title accordingly.

In addition to the title problem, it is also relevant from a biological point of view. Because the the model only assumes horizontal transmission, it clearly removed a key difference between sexual and asexual reproducers in respect of transmissible cancers. I would expect rare vertical transmission when individuals reproduce sexually and almost invariable vertical transmission during asexual reproduction, which could potentially turn this whole model upside down. I also believe that implementing vertical transmission in the model is key and would not be difficult (maybe I am wrong?): an offspring would acquire the transmissible cancer of the parent(s) if it has the same genotype at the interaction loci as the parent with transmissible cancer (only in case some sort of parental care (e.g. gestation/post-embryonic care) is presumable, due to physical proximity (maybe not even applicable if we talk about early multicellular organisms)). On the other hand, asexual reproduction would result in vertical transmission, maybe with a given rate, in hydra for instance almost all buds of cancerous hydra develop cancer.

Also the model assumes no evolution of anti-cancer mechanisms, which is another key difference between sexual and asexual reproducers. It is clearly articulated in numerous articles, that sexual reproduction accelerates adaptive evolution (see e.g. Goddard 2016 Nature), so once a transmissible cancer is present in the population sexual reproduction should accelerate the development of immunity/suppression. Even in early stages of such adaptations, sexually reproducing individuals might live a little bit longer than asexuals after acquiring cancer, which might result in their slightly higher fitness. There are indications that even Devil Facial Tumor Disease can accelerate selection in immune- and tumor-related genes, which could arguably be much slower in asexually reproducing species, e.g. in the case of Devils and DFTD, selection of defence mechanisms against the cancer have rapidly evolved in response to the epidemic (4-6 generations). Surely this selection would be much slower or perhaps not existing in the case asexual reproduction. This component might be harder to implement in the model, but should be clearly indicated in the ms as a shortfall of the work I believe. Perhaps authors should acknowledge this and suggest that the results should be interpreted with caution. If the model assumes no evolution of anticancer mechanisms then is difficult to put the results in the context of our paper.

I was a bit confused with the epidemiological model and how they modelled betta. In the supp material they say that betta increases the prevalence of transmissible cancers at equilibrium but is this fully independent of mortality and birth rates? Also, the birth rate and K have no effect on the proportion of cancers or selection, but surely these parameters are in reality somehow connected? This is just a thought as I had trouble with following the model, which surpassed my knowledge.

Some other minor clarifications needed:

Line 330 -333, they claim three cases of transmissible cancers have been identified in dogs, devils and bivalves. There are more than three transmissible cancers, I think they should say transmissible cancers have been found on xx species or alter the text according to the number of transmissible cancers they are referring to.

Lines 363 - 366. More precisely, in Tasmanian devil, a new lineage of transmissible cancer has been shown to express the host's major histocompatibility complex class 1 (MHC1) molecules. … Simultaneously, however, an older lineage of transmissible cancers in Tasmanian devils avoids the immune response by downregulating the expression of the MHC1 molecules (Devil Facial Tumour 1;Caldwell and Siddle, 2017).

In my opinion, DFT2 should not be defined as a new 'lineage' of transmissible cancer as implies that it evolved from DFTD, which is not the case. The manuscript first defines the devil cancer as DFTD, but then it uses DFT1 which is the same cancer. I know this might sound pedantic, but it would be good promote consistency in the devil literature because more and more papers are mixing the cancers and using them incorrectly. Ideally, they should explain the terminology early in the manuscript and keep consistency. 

Else congratulation for this nice piece of work!

Fred Thomas

Reviewer #3: In their manuscript the authors probe the theory that the evolution of sexual reproduction in multicellular organisms was driven by the emergence and spread of contagious cancers. Intuitively these cancers are similar to parasitic infection and as such the authors propose the hypothesis that the emergence and spread of these cancers initiates fluctuations in selection to drive the evolution of sexual reproduction, increasing genetic recombination. The authors initially test this hypothesis by building a population genetic model with an assumption that transmissibility is determined by a single locus defining self/non-self. This model predicts that fluctuations in selection are not significant enough to drive the emergence of sexual reproduction.

The population model is then supplemented with an epidemiological model to link the proportion of 'neocancers' and the selection coefficient. They are then able to vary a number of parameters to determine which may produce the outcome of sexual reproduction. Again, the emergence to sex is unlikely in most scenarios, with the exception of where the transmission rate is high and selection is high. 

The authors propose that the genetic link between newly emerged contagious cancers (so called neocancers) and their host is tight, which reduces selection and the 'Red Queen' scenario which would favour sexual reproduction.

The paper articulates an interesting hypothesis and builds the models logically to test this hypothesis. The primary issue with the models is their simplicity, numerous assumptions are made to simplify the model, such as the presence of a single locus determining self/non-self and an assumption of no mutation in tumour and host in the population model. These assumptions are obviously fundamental to the outcomes of the tests.

Central to this is the assumption that a single genetic locus determines self/non-self. Thus, the likelihood that the cancers will transmit to 'matched' individuals is very high and explains the genetic link between the host and cancer. The authors should introduce additional loci to the genetic model and test the result. This could be done in a stepwise manner to assess the impact on selection. 

Similarly, in figure 1 a three locus model and one locus model is tested (Figure 1A). There needs to be a clearer explanation of these two models and why one is favoured.

In the methods (line 96) the authors state that host and cancer cells form two large populations of constant size. This needs to be clarified, does this mean that the cancer emergence is at a rate that results in equal cell numbers between host and cancers. This would not seem likely.

Typo in line 150 - should read 'their sum is equal'.

Typo in line 170 - additional word - our

---

## [Decision Letter · Decision Letter 2]

3 Sep 2020

Dear Dr Aubier,

Thank you for submitting your revised Research Article entitled "Transmissible cancers and the evolution of sex under the Red Queen hypothesis" for publication in PLOS Biology. I have now obtained advice from the original reviewers and have discussed their comments with the Academic Editor. 

Based on the reviews, we will probably accept this manuscript for publication, assuming that you will modify the manuscript to address the remaining points raised by the reviewers. Please also make sure to address the data and other policy-related requests noted at the end of this email.

IMPORTANT:

a) Please address the remaining requests from the reviewers, including the shared concern about your use of the Red Queen hypothesis.

b) After discussion with the Academic Editor, we think that this article would be best published as a Short Report. As your paper is already quite concise, no re-formatting is needed, but please select "Short Reports" as the article type when you re-submit. We can also do this for you.

c) We believe that all the graphs in your figures can be generated using the code which you intend to deposit in Dryad. Please can you provide a reviewer link or password so that we can check this, AND clearly cite the Dryad URL in the relevant main and supplementary Figure legends?

We expect to receive your revised manuscript within two weeks. Your revisions should address the specific points made by each reviewer. In addition to the remaining revisions and before we will be able to formally accept your manuscript and consider it "in press", we also need to ensure that your article conforms to our guidelines. A member of our team will be in touch shortly with a set of requests. As we can't proceed until these requirements are met, your swift response will help prevent delays to publication.

- a cover letter that should detail your responses to any editorial requests, if applicable

*Copyediting*

*Published Peer Review History*

*Early Version*

Sincerely,

Roli Roberts

Senior Editor,

rroberts@plos.org,

PLOS Biology

REVIEWERS' COMMENTS:

Reviewer #1:

[identifies herself as Joanna Masel]

My concerns have been satisfactorily addressed. One small requested change re equating the Red Queen or parasite coevolution hypothesis with fluctuating selection dynamics is that Haldane 1949 should not be cited in support of FSD. The FSD hypothesis began I believe with the Hamilton paper cited, and I don't think it is appropriate to back-project it onto earlier papers about host-parasite coevolution more generally. 

Optionally on a linguistic note, I know it is common practice to equate "Red Queen" with FSD, but I do not believe it is appropriate given the far broader definition of Red Queen in the original van Valen paleontology paper, concerning extinction rates, that I believe first introduced the term Red Queen to evolutionary biology. Unfortunately, I think the effect of this linguistic practice is to help crowd out alternative and more realistic theories of host-parasite coevolution. This is a pet peeve of mine that is not at all specific to the manuscript at hand, which has been satisfactorily revised.

Signed,

Joanna Masel

Reviewer #2:

[identifies himself as Frederic Thomas]

This version is really much improved, I love it. However, I still have two comments (the second one being really important in my opinion):

1- I think there is a relevant paper that should be mentioned in your discussion 

Khosrotehrani, K. et al. (2005) Natural history of fetal cell microchimerism during and following murine pregnancy. Journal of Reproductive Immunology DOI: 10.1016/j.jri.2005.02.001

This paper in fact simulates our scenario using syngenic mice. Indeed, using syngenic mice these authors mimicked an asexual reproduction in mice, and they showed that when embryos are identical to their mother, microchimerism is significantly enhanced. They are not using cancerous cells, I agree, but this findings really supports our idea that vertical cell transmission is more difficult to control and/or to prevent when parent and offsprings are identical. 

2- I still think that there is a problem in the title. You wrote:

"In this paper, we investigate whether antagonistic interactions between hosts and transmissible cancers can promote the evolution of sex under the Red Queen hypothesis. We analyze a population genetic model of fluctuating selection and complement it with an epidemiological model. The latter model builds an explicit epidemiological setting that we then use to examine the likely parameter values that the population genetic model takes. This combined use of two models allows us to evaluate how likely it is for the modelled system to find itself within a selection regime where Red Queen dynamics can favour sexual reproduction."

 The basic of the assumption for our paper was that transmissible cheater/cancer cells evolved at the same time as multicellularity. Due to their clonal reproduction the multicellular host cells could not recognise the transmissible cheater/cancer cells. Consequently, sexual reproduction evolved as an adaptive trait to prevent horizontal and/or vertical transmission of cheater/cancer cells. This was a one off evolutionary adaptation and hence once it appeared no further adaptations occurred. Importantly, our scenario should therefore not be regarded as a red-queen process, which is a constant, and ongoing battle between host and pathogens. 

Signed: Fred Thomas

---

## [Editor Report · Decision Letter 3]

21 Sep 2020

Dear Dr Aubier,

On behalf of my colleagues and the Academic Editor, Nick H Barton, I am pleased to inform you that we will be delighted to publish your Short Reports in PLOS Biology. 

Early Version

PRESS 

Kind regards,

Alice Musson

Publishing Editor, 

PLOS Biology

on behalf of

Roland Roberts,

Senior Editor

PLOS Biology